# Distribution of Acetogenic Naphthoquinones in Droseraceae and Their Chemotaxonomic Utility

**DOI:** 10.3390/biology13020097

**Published:** 2024-02-03

**Authors:** Jan Schlauer, Andreas Fleischmann, Siegfried R. H. Hartmeyer, Irmgard Hartmeyer, Heiko Rischer

**Affiliations:** 1The Center for Plant Molecular Biology (ZMBP), University of Tuebingen, Auf der Morgenstelle 32, D-72076 Tuebingen, Germany; 2Botanische Staatssammlung München, Menzinger Strasse 67, D-80638 Munich, Germany; fleischmann@bio.lmu.de; 3GeoBio-Center LMU, Ludwig-Maximilians-University Munich, D-80539 München, Germany; 4Independent Researcher, Wittlinger Str. 5, D-79576 Weil am Rhein, Germany; s.hartmeyer@t-online.de (S.R.H.H.); irmgard@hartmeyer.de (I.H.); 5VTT Technical Research Centre of Finland Ltd., Tekniikantie 21, FIN-02150 Espoo, Finland

**Keywords:** chemotaxonomy, phytochemistry, acetogenic naphthoquinones, plumbagin, ramentaceone, Droseraceae, *Drosera*, *Dionaea*, *Aldrovanda*, carnivorous plants

## Abstract

**Simple Summary:**

Recent research and chemical screening work has identified the distribution of two major naphthoquinone regioisomers across a representative selection (including all genera, all sections in the large genus *Drosera*, and 190 of ca. 260 species recognized to date) in the diverse carnivorous plant family Droseraceae (sundews, Venus’s flytrap and waterwheel plant). A consistent and reliably reproducible foundation and framework for the systematic classification and phylogenetic evaluation of mutual relationships is thus provided. This review gives an updated overview of the naphthoquinone data relevant to the chemotaxonomy of the group under investigation. The essential conclusions from these data are summarized and directions are proposed for future research.

**Abstract:**

Chemotaxonomy is the link between the state of the art in analytical chemistry and the systematic classification and phylogenetic analysis of biota. Although the characteristic secondary metabolites from diverse biotic sources have been used in pharmacology and biological systematics since the dawn of mankind, only comparatively recently established reproducible methods have allowed the precise identification and distinction of structurally similar compounds. Reliable, rapid screening methods like TLC (Thin Layer Chromatography) can be used to investigate sufficiently large numbers of samples for chemotaxonomic purposes. Using distribution patterns of mutually exclusive naphthoquinones, it is demonstrated in this review how a simple set of chemical data from a representative sample of closely related species in the sundew family (Droseraceae, Nepenthales) provides taxonomically and phylogenetically informative signal within the investigated group and beyond.

## 1. Introduction

Naphthoquinones are an important class of natural products (Figure 1) containing pigments (e.g., alkannin, shikonin, lawsone), allelochemicals (juglone), cytostatics (lapachol, dunnione, plumbagin, chimaphilin), phytoalexins (mansonones), and vitamins (phylloquinone) [1,2]. At least five distinct biosynthetic routes (several of which established several times independently) are known to yield these bicyclic metabolites [3]: The acetate-polymalonate or polyketide pathway (plumbagin, ramentaceone), a versatile and widespread route commonly used to produce aromatic compounds characterized by oxygen substitution at every second atom of the carbon backbone that is even retained in metabolites of mixed biosynthetic origin (like, e.g., flavonoids and stilbenoids); The *o*-succinylbenzoate pathway (juglone, lawsone, lapachol, phylloquinone, dunnione), a route that is almost ubiquitous in plants, as it leads to the essential phylloquinone (vitamin K group);The homogentisate/mevalonate pathway (chimaphilin) so far known only from Ericaceae-Pyroloideae to produce chimaphilin and its immediate derivatives;The 4-hydroxybenzoate/geranyl-pyrophosphate pathway (shikonin, alkannin), known with certainty only from Boraginaceae;The farnesyl-pyrophosphate pathway (mansonones) common to the huge and widespread class of sesquiterpenoids, of which only a comparatively limited number become quinones by oxidation.

Nevertheless, there are only a few cases in which a specific naphthoquinone structure is formed convergently via different routes.

The regioisomers plumbagin (2-methyljuglone) and ramentaceone (7-methyljuglone) are characteristic naphthoquinones within the angiosperm order Nepenthales or “non-core Caryophyllales” (recorded so far in families Plumbaginaceae, Nepenthaceae, Droseraceae, Drosophyllaceae, Dioncophyllaceae and Ancistrocladaceae, Figure 2). As these 1,4-naphthoquinones demonstrably possess biological activities [1,2], preparations from sundew species (*Droserae* herba) used to be listed in several pharmacopoeias and are currently evaluated by European Pharmacopoeia standards [4]. Structurally and biosynthetically similar anthraquinones (e.g., emodin) are known from Polygonaceae within the same order [5]. Apart from genetic evidence, these characteristic metabolites clearly support the separation of Nepenthales from the sister order Caryophyllales (sensu stricto), which is in turn characterized by the common presence of betalains [6] that are lacking in Nepenthales [7,8]. Another noteworthy coincidence is the presence of at least six genera of carnivorous plants in four families of Nepenthales (*Nepenthes* of Nepenthaceae, *Dionaea*, *Aldrovanda* and *Drosera* of Droseraceae, *Drosophyllum* of Drosophyllaceae, *Triphyophyllum* of Dioncophyllaceae; carnivory is monophyletic in Nepenthales and became lost at least twice in this lineage, in two out of the three genera of Dioncophyllaceae and in all Ancistrocladaceae), while no carnivorous members of Caryophyllales are known [9].

Outside Nepenthales, noteworthy records of plumbagin and/or ramentaceone are limited to the phylogenetically distant families Ebenaceae (Ericales) and Iridaceae (Asparagales) [3]. Due to their indicative substitution patterns, it is assumed that the characteristic naphthoquinones in Nepenthales, Ebenaceae and Iridaceae are formed via the acetate-polymalonate route with hexaketide and 1,8-dihydroxy-3-methylnaphthalene intermediates [10]. A number of biosynthetic precursors, structural variants and di- or polymers have been described (e.g., droserone, isoshinanolone, shinanolone, 7-methylnaphthazarin, dihydroplumbagin, maritinone, mamegakinone [1,11], Figure 3). Because: 1. their structural affinities to the respective naphthoquinone are obvious; 2. they usually co-occur with the same regioisomer; and 3. their distribution has been portrayed satisfactorily before [1,11], the additional presence of one or several of these derivatives is of less chemotaxonomic interest here. Thus, they will not be addressed in detail in this review that will instead focus on plumbagin and ramentaceone in the Droseraceae. It should, however, be noted that naphthaldehydes and their polymers appear to be particularly common in Ebenaceae only, where they may contribute to the coloration of the heartwood (ebony, [12,13]). 

Droseraceae are among the most diverse families of carnivorous plants, containing three genera, of which two are monotypic (*Aldrovanda*, a rare but widely distributed-azonal-aquatic in the Old World, and *Dionaea*, endemic to few localities in the southeastern United States). The third genus, *Drosera*, contains about 260 described species of almost cosmopolitan distribution but with only a few representatives (essentially from just a single section) in the northern hemisphere. A certain degree of motility in the leaf lamina and/or its substructures is usually associated with the adhesive (*Drosera*) or snap trap (*Aldrovanda*, *Dionaea*) mechanisms for prey capture. This is unique within carnivorous Nepenthales, of which the other members use passive (pitcher or adhesive) trap types [9].

Throughout the past decade, significant progress has been made in the screening for naphthoquinones in Droseraceae [14,15,16,17,18,19,20,21,22], and although several papers aimed to address these specific metabolites at various degrees of accuracy [1,23,24,25], none of the recent reviews represented their meanwhile known isomer distribution satisfactorily. It is the aim of the present account to fill this gap, so that chemotaxonomic considerations can be based on adequate data.

## 2. Review of Naphthoquinone Data from Droseraceae

The distribution of individual naphthoquinone isomers that have been detected in identified species of Droseraceae is summarized in Table 1 that also contains information on the classification of the investigated taxa, the provenance of the investigated samples and comments that essentially relate to the (few) differences between subsequent, independent studies.

## 3. Discussion

### 3.1. Screening Approaches to Naphthoquinones in Droseraceae

Since the first comparative study that covered a representative selection of (19) species of Droseraceae [26], Thin Layer Chromatography (TLC) has been established as a simple yet reliable, sensitive and highly specific detection and separation method for plumbagin and ramentaceone. This has been confirmed convincingly by a subsequent, more inclusive (63 species) survey that already outlined major distribution trends for either main quinone in several of the larger lineages within *Drosera* [27]. A recent review on secondary metabolites from Nepenthales [25] summarized “investigations based on modern identification techniques” to the explicit and deliberate exclusion of results from TLC, arguably the most efficient tool ever applied in comparative studies on these same metabolites. While more detailed analyses using additional methods, such as High-Performance Liquid Chromatography (HPLC) or Ultra-High-Performance Liquid Chromatography (UPLC) with Mass Spectrometry (MS) and Nuclear Magnetic Resonance spectroscopy (NMR), have indeed contributed to the detection and structural elucidation of several derivatives and potential biosynthetic intermediates of plumbagin and ramentaceone [11], TLC is and remains the principal source of secondary metabolite distribution data throughout Nepenthales and in Droseraceae.

### 3.2. Role of Naphthoquinones in Carnivory

While naphthoquinones may aid in preserving animal prey and protecting plant traps during digestion [44], and precursors acquired by carnivory are channeled into their biosynthetic pathway [45], the speculation that this may be an “important adaptation” across all carnivorous Nepenthales [46] is flawed by the fact that there are a few species of *Nepenthes* [47] and a comparatively large number of species of *Drosera* (Table 1, [21,22,27]) in which neither plumbagin nor ramentaceone have been detected. Although it was mentioned that the acetogenic naphthoquinones of carnivorous Nepenthales are lacking in all other lineages of carnivorous plants [46] where, interestingly, other characteristic acetogenic polyketides occur in some families such as the alkaloid coniine in *Sarracenia* [48], it was not appropriately considered that plumbagin is present in the closely related but non-carnivorous Plumbaginaceae (consecutive sister to the carnivorous clade of Nepenthales [9]) and Ancistrocladaceae (carnivory lost secondarily [9]). Both plumbagin and ramentaceone are known outside Nepenthales from the likewise non-carnivorous but phylogenetically distant Ebenaceae and Iridaceae [1], where they can obviously not constitute any adaptation to carnivory.

Carnivory has obviously been lost in most Dioncophyllaceae and throughout Ancistrocladaceae [9]. However, even these families retained the ability to form acetogenic naphthoquinones and/or tetralones [1,49], and they have additionally acquired the ability to form naphthylisoquinoline alkaloids (e.g., dioncophylline A, Figure 3), evidently from common biosynthetic precursors [50]. The erroneous statements that there “are no reports on the biosynthesis of this type of alkaloid” and “the origin of these alkaloids is phenylalanine” [51] are obviously untenable and incomprehensible in the light of the quite extensive relevant literature [50] that has been outright ignored.

### 3.3. Taxonomic Implications

#### 3.3.1. Quinone Relationships of Droseraceae and Nepenthales

Three biosynthetic routes towards naphthoquinones are specific to a few angiosperm orders phylogenetically distant from Nepenthales, viz the 4-hydroxybenzoate/geranyl-pyrophosphate pathway leading to shikonin and alkannin in Boraginaceae (Boraginales), the homogentisate/mevalonate route to yield chimaphilin that is unique to Ericaceae subfamily Pyroloideae (Ericales), and the farnesyl-pyrophosphate pathway to mansonones and related cadinane sesquiterpenoids in Malvaceae (Malvales), Aristolochiaceae (Piperales), Ulmaceae (Rosales), Zingiberaceae (Zingiberales) [1,3,52,53].

Additionally, the geranylgeranyl-pyrophosphate pathway is used to form serrulatane diterpenoids (“extended cadinanes”) that may be converted to naphthoquinones (e.g., biflorin) in a similar fashion in Scrophulariaceae and Plantaginaceae (both Lamiales) [54]. These are noteworthy examples of convergence in chemical structure [3] without a common phylogenetic or biochemical background.

Several further naphthoquinones of unknown or unconfirmed biosynthetic origin [3] or of limited taxonomic distribution are scattered across the plant kingdom [1]; therefore, little can be said about their taxonomic significance at present.

Interestingly, a mutual exclusivity of emodin type octaketides and plumbagin type hexaketides is found both in Nepenthales (see below) and in the monocot order Asparagales, where the octaketide quinones are confined to Asphodelaceae (incl. Aloaceae, Dianellaceae, Eccremidaceae, Geitonoplesiaceae, Hemerocallidaceae, Johnsoniaceae, Phormiaceae, Xanthorrhoeaceae) [5], while hexaketide quinones are confined to Iridaceae [1,3]. This type of chemical similarity differs from the examples above because within the orders a common biosynthetic route (in this case, the acetate-polymalonate pathway) leads to quinone structures of different size but still retains characteristics that suggest phylogenetic relationship; thus, it does not appear to be convergent. Between the distant orders, the dualism between octaketides and hexaketides may indicate similar selective pressures that have resulted in similar developments; thus, the dualism as such is supposedly convergent. The genetics of these polyketides is, however, only poorly understood, and beyond a few expectable enzymes catalyzing individual biosynthetic steps [10], no comparative sequence data are known to reflect the obvious metabolic similarities nor the anticipated genetic differences.

No dualism between different polyketide classes is known with certainty in Ericales, where Ebenaceae stand isolated with their hexaketide-derived naphthoquinones [1,3]. Occasional reports of emodin from Actinidiaceae, Balsaminaceae and Primulaceae subfam. Myrsinoideae [5] are questionable and require confirmation of both the chemical structures and the biosynthetic routes. Ericales is nevertheless the order with the highest known and confirmed naphthoquinone diversity, as it also includes Ericaceae subfam. Pyroloideae containing the unique chimaphilin (see above), and Balsaminaceae containing lawsone (also known from Lythraceae, Myrtales). The latter is produced from *o*-succinylbenzoate [1,3], a common intermediate among plants because it also leads to phylloquinone, to juglone in Juglandaceae (Fagales) [1,3], and to the characteristic naphtho- and anthraquinones in numerous families of Lamiales (e.g., dunnione and lapachol), Proteaceae (Proteales, e.g., lomatiol), and of Rubiaceae (Gentianales, predominantly alizarin type anthraquinones) that are *o*-succinylbenzoate derivatives with a hemiterpene substituent at the position that is occupied by a diterpene chain in phylloquinone [2].

In Nepenthales, the early branching families Tamaricaceae and Frankeniaceae are not known to yield any characteristic quinones [1,5].

Polygonaceae is the only family in Nepenthales characterized by various octaketides like the anthraquinone emodin [5] instead of hexaketide-derived naphthoquinones.

In Plumbaginaceae, possibly the sister family of Polygonaceae, plumbagin has been recorded in all investigated species of subfam. Plumbaginoideae [1,55,56]. Ramentaceone has not been found yet in Plumbaginaceae, and none of the isomers are known so far from the subfam. Staticoideae [1].

In Ancistrocladaceae, Dioncophyllaceae and Drosophyllaceae, plumbagin and several of its derivatives and/or potential biosynthetic precursors have been detected [1,49,57]. This strongly indicates that plumbagin formation is a symplesiomorphic character state among those members of Nepenthales that contain acetogenic naphthoquinones.

In *Drosophyllum* (Drosophyllaceae), dihydroramentaceone has been identified as an additional constituent besides plumbagin [11]. A previous report of ramentaceone itself [27] was not confirmed afterwards but it may possibly be formed spontaneously from the dihydroquinone.

The tendency that both monotypic genera (*Aldrovanda*, *Dionaea*) of Droseraceae and the earliest branching lineage in *Drosera* (*D. regia*) contain plumbagin (Table 1, Figure 4) confirms the general trend that this isomer is also more widespread outside Droseraceae. Within Nepenthales, the only other family in which a reasonable number of species have been identified to contain the other isomer (ramentaceone) is Nepenthaceae [47], the immediate sister of Droseraceae [9].

#### 3.3.2. Quinone Relationships within Drosera

The different sections in the predominantly Australian *Drosera* subgenus *Ergaleium* show noteworthy correlations between quinone patterns and life-forms.

*D.* sect. *Coelophylla* (a single species of therophytes confined to southwestern and southern Australia) contains ramentaceone and is thus phytochemically characterized as a distinct, supposedly early branching clade within the subgenus (supported phylogenetically [43]). Additionally, the section also features a unique, one-shot catapult-flypaper trap [58].

The three morphologically (but not phylogenetically) separated groups *Ergaleium*, *Stoloniferae*, and *Erythrorhiza* of *D.* sect. *Ergaleium* are composed of stem tuber geophytes, and most of which are confined to southwestern Australia. The delimitation of the three subgroups is based on morphological characteristics, of which some may be convergent; thus, the assignment of a few species in Table 1 is tentative. In this group plumbagin is usually the dominant naphthoquinone. The detection of ramentaceone in two taxa of *D*. sect. *Stoloniferae* indicates that screening this group further will yield taxonomic information.

With plumbagin as the main quinone *D*. sect., *Phycopsis* (hemicryptophytes from E Australia to New Zealand) is chemically close to the tuberous sections.

Sections *Lasiocephala* and *Bryastrum* are largely hemicryptophytes; a few species of the former are also bulbous geophytes. While *D*. sect. *Bryastrum* is almost confined to SW Australia (but two species, one, *D. pygmaea* extending to southern Australia and NZ, and the other, *D*. *meristocaulis*, has a disjunct distribution confined to a single mountain peak in the Amazonian part of the Guayana Highlands); *D*. sect. *Lasiocephala* is distributed from northern Australia to New Guinea. In the *D*. sect., transformed *Bryastrum* leaves (gemmae) are seasonally formed for vegetative reproduction, while gemmae are missing in *D*. sect. *Lasiocephala* (and in *D*. *meristocaulis*). Although the two groups are morphologically, ecologically and chorologically distinct, both sections are phylogenetic sister lineages and are notable for the absence of naphthoquinones in most species (including the disjunct *D*. *meristocaulis*), clearly a secondary loss. The presence of a few species that contain quinones in both sections indicates that the loss has either occurred twice independently or that quinone production is easily restored.

The other large subgenus, *D*. subgen. *Drosera* is far more widespread (subcosmopolitan) and is composed predominantly of hemicryptophytes adapted to various climatic conditions.

*D*. sect. *Thelocalyx* has a tropical distribution in Australasia and South America. Its two species are therophytes that are so similar to each other that it can become difficult to identify them if the provenance is unknown. This is also reflected in their common formation of ramentaceone.

*D*. sect. *Prolifera* is another tropical group but this one is restricted to northern Queensland. Ramentaceone is found in this section as well, but *D*. *prolifera* contains plumbagin instead.

*D*. sect. *Arachnopus* consists of therophytes with one species from Africa to Eastern Asia, three species in SE Asia and Australia and about a dozen endemic to Australia. Besides the adaptation to an annual life cycle, this section has developed an amazing diversity of quinone patterns. It is noteworthy that morphologically similar and geographically overlapping sister species (*D*. *aquatica*/*D*. *nana* and *D*. *hartmeyerorum*/*D*. *barrettiorum*) usually also exhibit similar quinone patterns (ramentaceone dominant in the mentioned species). The widespread *D*. *indica* contains both quinone isomers, which is a rare feature in *Drosera*, usually resulting from hybridization [14].

*D*. sect. *Stelogyne*, a monotypic section comprising a single relict species from SW Australia containing ramentaceone (like in the more speciose sections mentioned below), occupies a morphologically and phylogenetically isolated position in *D*. subgen. *Drosera*.

*D*. sect. *Psychophila* has a “Gondwanan” distribution with one species in New Zealand and the other from southern South America (cf. the decidedly more tropical *D*. sect. *Thelocalyx* that likewise has a gap in Africa). Their quinone patterns involve both isomers at various proportions and do not provide clear taxonomic clues.

*D*. sect. *Drosera* is distributed throughout the northern hemisphere and in South America, with four species extending into tropical Asia and Malesia and one of them further to Australia and New Zealand. The section is characterized by a dominance of ramentaceone in most species. Most members of this lineage are diploid. Where hybridization involves the few species of this sections that contain plumbagin, the hybrids usually contain both isomers [14].

Essentially the same situation (ramentaceone dominant) is also present in *D*. sect. *Ptycnostigma* (Africa) and in *D*. sect. *Brasilianae* (Brazil)—both sections largely comprise tetraploids or polyploids. *D*. sect. *Brasilianae* shows an additional trend by several species lacking naphthoquinones whatsoever, which may be another (and very probably independent) secondary loss involving several related species in the genus.

## 4. Conclusions

As detailed in the discussion, the acetogenic naphthoquinones plumbagin and ramentaceone are valuable phytochemical markers that aid in the delimitation, identification and classification of taxa in Droseraceae (and beyond). Their presence/absence is stable enough within and between populations and taxa to provide reliable taxonomic information and phylogenetic signal.

In comparison to other groups of comparable distribution and size, plants of the order Nepenthales belonging to other families than Droseraceae have been subject to rather thorough investigation with respect to their secondary metabolites [1,47,49,55,56,57]. However, whereas the naphthylisoquinoline alkaloids of Ancistrocladaceae are fairly well known [59], only three of 21 species of *Ancistrocladus* have been investigated for naphthoquinones so far [1].

A few species groups in *Drosera* are still underinvestigated, e.g., *D*. subgen. *Arcturia*, *D*. sect. *Ergaleium*—“*Stoloniferae*”, *D*. sect. *Prolifera*, and, to some extent, *D*. sect. *Arachnopus*. Access to suitable plant material for investigation is the main bottleneck for chemotaxonomic progress that can be expected in these cases.

## Figures and Tables

**Figure 1 biology-13-00097-f001:**
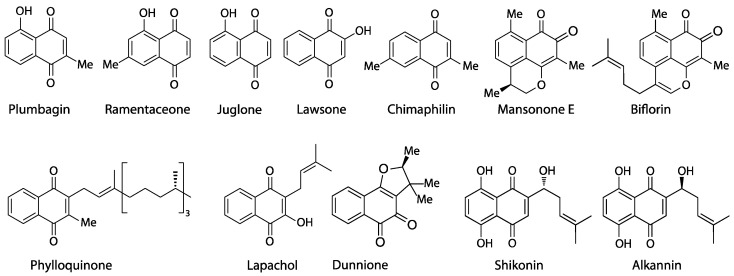
Representative natural naphthoquinones.

**Figure 2 biology-13-00097-f002:**
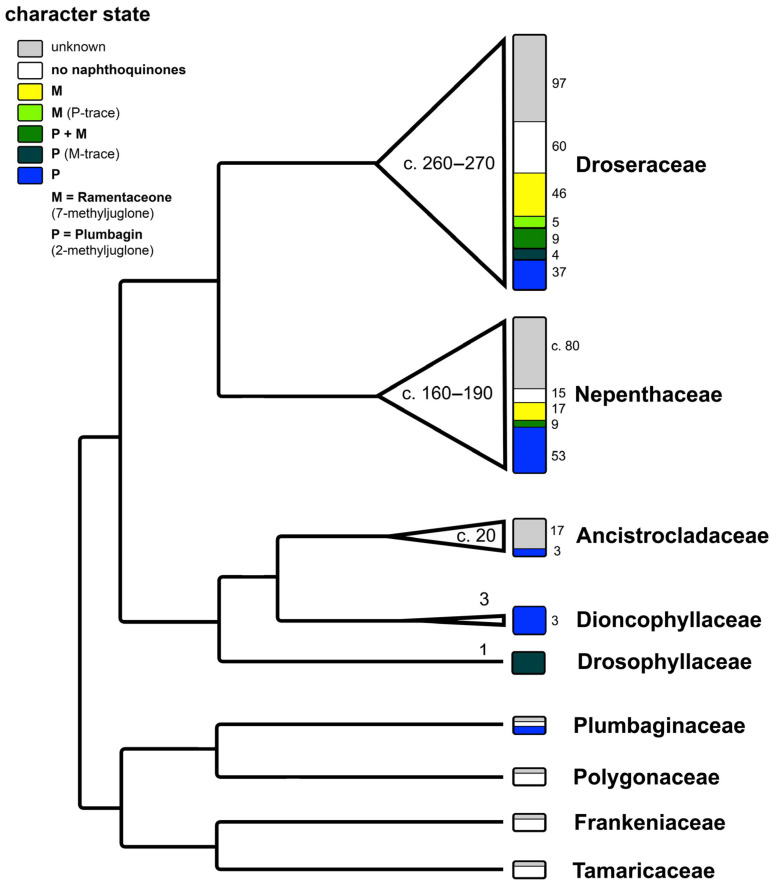
Phylogenetic pattern of naphthoquinone distribution in Nepenthales. Cladogram modified from [9]. Numbers within the triangles and width of the triangles correspond to species number of each lineage. Numbers in small font correspond to the number of species showing the respective character state. Illustration by Andreas Fleischmann.

**Figure 3 biology-13-00097-f003:**
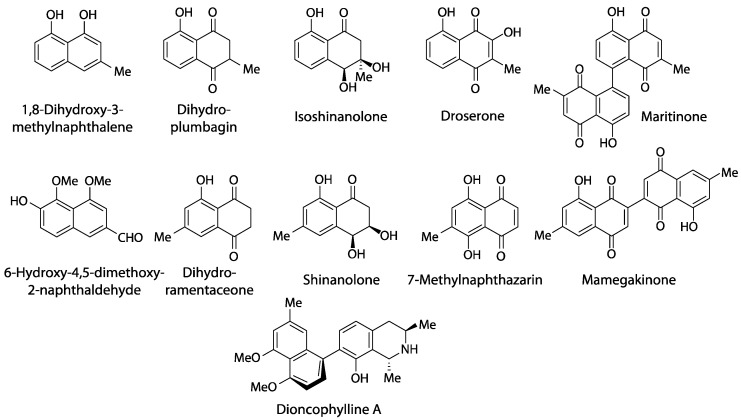
Naphthalene derivatives related to plumbagin and ramentaceone.

**Figure 4 biology-13-00097-f004:**
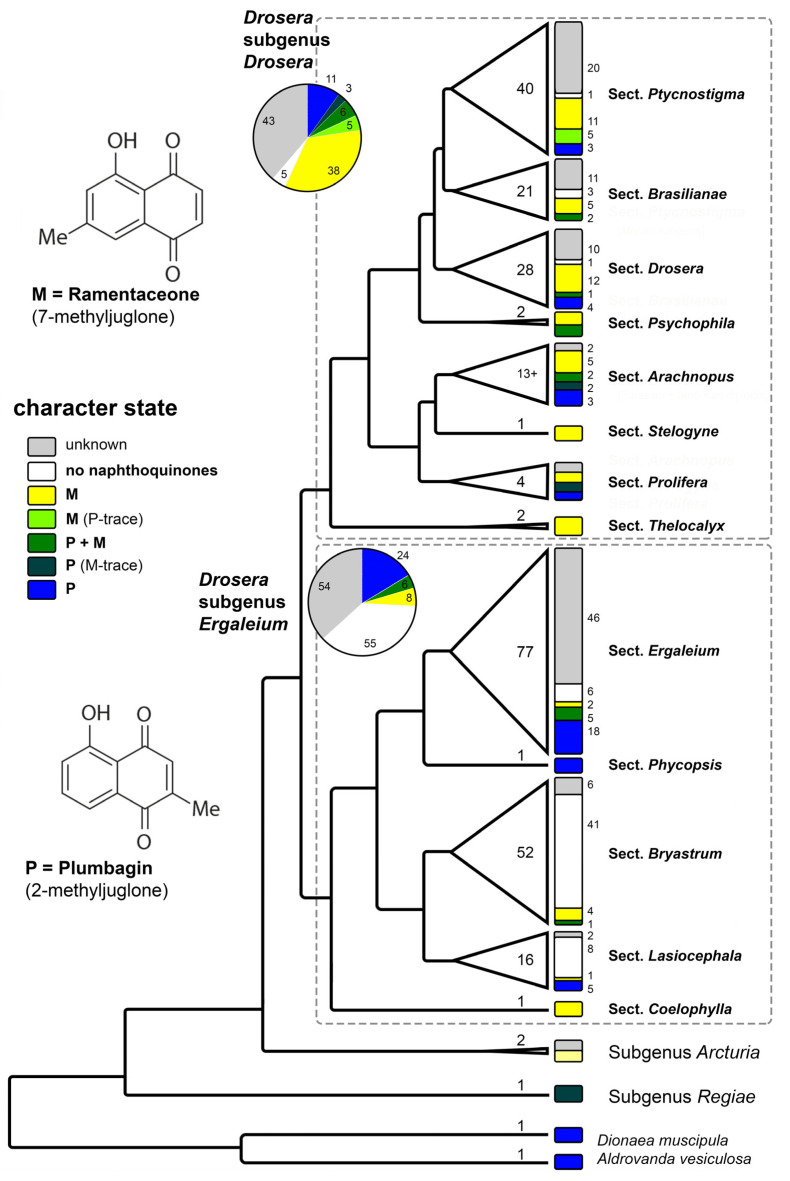
Phylogenetic pattern of naphthoquinone distribution in Droseraceae. Cladogram modified from [43]. Numbers within the triangles and width of the triangles correspond to species number of each lineage. Numbers in small font correspond to the number of species showing the respective character state. For this diagram, only measurements from [11,14,15,16,17,18,19,20,21,22] were considered. Illustration by Andreas Fleischmann.

**Table 1 biology-13-00097-t001:** Naphthoquinone distribution and classification of Droseraceae taxa.

Classification ^1^	Provenance ^2^	Taxon	Quinones ^3^	References	Comments
*Dionaea*	SE USA	*D. muscipula*	P	[26,27]	
*Aldrovanda*	Old World	*A. vesiculosa*	P	[26,27]	
*Drosera* subg. *Regiae*	ZA	*D. regia*	P, (M-tr)	[18,27]	
*Drosera* subg. *Arcturia*	Tasmania	*D. arcturi*	(M-tr)	[18]	0 in [27]
*Drosera* subg. *Ergaleium*					
*Drosera* sect. *Coelophylla*	SW AU	*D. glanduligera*	M	[15]	
*Drosera* sect. *Lasiocephala*	N AU, N.Guin.	*D. banksii*	(P-tr)	[22]	
	NT	*D. brevicornis*	0	[17,22]	
	NT	*D. broomensis*	P	[22]	
	N AU	*D. caduca*	P	[22]	
	Kimberley	*D. darwinensis*	0	[22]	
	Kimberley	*D. derbyensis*	P	[22]	
	Kimberley, NT	*D. dilatatopetiolaris*	0	[22]	
	NT	*D. falconeri*	0	[22]	
	NT	*D. fulva*	M	[22]	
	Kimberley, NT	*D. kenneallyi*	P	[22]	
	Qld.	*D. lanata*	0	[22]	
	NT	*D. aff. lanata*	0	[22]	
	NT	*D. aff. paradoxa* (“swamp form”)	0	[17,22]	
	NT	*D. aff. paradoxa* (“NT form”)	0	[22]	as *D. paradoxa* “type form”
	NT	*D. aff. paradoxa* (orange form)	0	[22]	
	NT	*D.* cf. *petiolaris*	0	[22]	
	Qld.	*D. petiolaris*	0	[22]	
	N AU, N.Guin.	“*D. petiolaris*”	0	[27]	
*Drosera* sect. *Bryastrum*	SW AU	*D. androsacea*	0	[21]	
	SW AU	*D. australis*	0	[21]	
	SW AU	*D. barbigera*	0	[27]	
	SW AU	*D. bindoon*	0	[21]	
	SW AU	*D. callistos*	0	[21]	
	SW AU	*D. citrina*	0	[21]	
	SW AU	*D. closterostigma*	0	[21]	
	SW AU	*D. coomallo*	0	[21]	
	SW AU	*D. depauperata*	0	[21]	
	SW AU	*D. echinoblastus*	0	[21]	
	SW AU	*D. enneabba*	0	[21]	
	SW AU	*D. gibsonii*	0	[21]	
	SW AU	*D. grievei*	0	[21]	
	SW AU	*D. helodes*	0	[21]	
	SW AU	*D. hyperostigma*	0	[21]	
	SW AU	*D. lasiantha*	0	[21]	
	SW AU	*D. leucoblasta*	0	[27]	
	SW AU	*D. leucostigma*	0	[21]	
	SW AU	*D. mannii*	0	[21]	
	BR, Venezuela	*D. meristocaulis*	0	[28]	
	SW AU	*D. micrantha*	0	[21]	
	SW AU	*D. microscapa*	0	[21]	
	SW AU	*D. miniata*	0	[21]	
	SW AU	*D. minutiflora*	M	[21]	
	SW AU	*D. nitidula*	0	[27]	
	SW AU	*D. nivea*	0	[21]	
	SW AU	*D. occidentalis*	0	[27]	
	SW AU	*D. omissa*	M	[21]	0 in [27]
	SW AU	*D. oreopodion*	0	[21]	
	SW AU	*D. paleacea*	0	[21,27]	
	SW AU	*D. parvula*	0	[27]	
	SW AU	*D. patens*	0	[21]	
	SW AU	*D. pedicellaris*	M	[21]	
	SW AU	*D. platystigma*	0	[27]	
	SW AU	*D. pulchella*	P + M	[21]	0 in [27]
	SW AU	*D. pulchella* × *nitidula*	0	[27]	
	SW AU	*D. pulchella* × *occidentalis*	0	[27]	
	SW AU	*D. pycnoblasta*	0	[27]	
	E AU	*D. pygmaea*	P	[29]	most probably in err.
	E AU	*D. pygmaea*	0	[27]	
	SW AU	*D. roseana*	0	[21]	
	SW AU	*D. sargentii*	M	[21]	
	SW AU	*D. scorpioides*	0	[27]	
	SW AU	*D. sewelliae*	0	[21]	
	SW AU	*D. silvicola*	0	[21]	
	SW AU	*D. stelliflora*	0	[21]	
	SW AU	*D. trichocaulis*	0	[21]	
	SW AU	*D. verrucata*	0	[21,27]	(*D. dichrosepala* auct. in err.)
	SW AU	*D. walyunga*	0	[21]	
*Drosera* sect. *Phycopsis*	E AU	*D. binata*	P	[26,27,30,31,32,33]	
	E AU	*D. binata* (var. *dichotoma*)	P	[26,30,33]	
*Drosera* sect. *Ergaleium—*“*Ergaleium*”	SW AU	*D. andersoniana*	P	[27]	
	E AU	*D. auriculata*	P, (M-tr)	[26,27,30,33]	
	SE AU	*D. auriculata*	P	[22]	
	SW AU	*D. gigantea*	P	[33,34]	
	N.Guin.	*D. gracilis*	P, (M-tr)	[22]	
	E AU	*D. gunniana*	P, (M-tr)	[22]	
	SE AU	*D. hookeri*	P, (M-tr)	[22]	
	Thailand	*D. lunata*	P + M	[22]	
	E Asia	*D. lunata*	P	[26,27,30,33,35,36]	
	SW AU	*D. macrantha s. str.*	P	[22]	0 in [27]
	SW AU	*D. marchantii*	0	[27]	
	SW AU	*D. menziesii*	P	[17]	0 in [27]
	SW AU	*D. microphylla*	P	[27,30]	
	SW AU	*D. modesta*	P	[17,27]	
	SW AU	*D. moorei*	(P-tr)	[22]	
	SW AU	*D. myriantha*	0	[27]	
	SW AU	*D. neesii*	0	[27]	
	SW AU	*D. planchonii*	P	[17]	
	SW AU	*D. radicans*	0	[27]	
	SW AU	*D. subhirtella*	0	[27]	
	SW AU	*D. zigzagia*	P	[22]	
*Drosera* sect. *Ergaleium—*“*Erythrorhiza*”	SW AU	*D. aberrans*	P	[17]	
	SW AU	*D. aberrans*	P + M	[22]	
	SW AU	*D. bulbosa*	P	[27]	
	SW AU	*D. erythrorhiza*	P	[27,30]	
	SW AU	*D. lowriei*	P	[17]	
	SW AU	*D. macrophylla*	P	[27]	
	SW AU	*D. major*	0	[17]	
	SW AU	*D. tubaestylis*	P	[17]	
	SW AU	*D. whittakeri*	P	[26,27,30,31,33]	
*Drosera* sect. *Ergaleium—*“*Stoloniferae*”	SW AU	*D. platypoda*	P	[27]	
	SW AU	*D. rupicola*	M	[17]	P in [27]
	SW AU	*D. stolonifera*	M	[27,30]	0 in [22]
*Drosera* subg. *Drosera*					
*Drosera* sect. *Thelocalyx*	NT, India	*D. burmannii*	M	[16,27]	P in [33,37], probably in err. (method does not identify isomer)
	BR	*D. sessilifolia*	M	[16]	
*Drosera* sect. *Prolifera*	Qld.	*D. adelae*	(P-tr), M	[19,27,32,33]	
	Qld.	*D. prolifera*	P	[19,27,32,33]	
	cult. (Qld.)	*D. prolifera* × *schizandra*	P + M	[19]	
	Qld.	*D. schizandra*	M	[19]	0 in [27]
*Drosera* sect. *Arachnopus*	NT	*D. aquatica*	M	[15]	
	Kimberley	*D. aurantiaca*	M	[16]	
	Kimberley	*D. barrettiorum*	M	[22]	
	Kimberley	*D. cucullata*	P	[15,16]	
	NT	*D. finlaysoniana*	P, (M-tr)	[15,16]	
	NT	*D. fragrans*	P	[15]	
	Kimberley	*D. hartmeyerorum*	M	[15,16]	
	?AU	“*D. indica*”	P	[26]	“*D. indica*” included all spp. of *D*. sect. *Arachnopus* at time of study
	Vietnam, Ivory Coast	*D. indica*	P + M	[16,27]	
	Japan	*D. makinoi*	P + M	[20]	
	Kimberley	*D. margaritacea*	P, (M-tr)	[22]	
	NT	*D. nana*	M	[17]	
	NT, Qld., Japan	*D. serpens*	P	[15,16,20]	
*Drosera* sect. *Stelogyne*	SW AU	*D. hamiltonii*	M	[18,26,27,32,33]	
*Drosera* sect. *Psychophila*	New Zealand	*D. stenopetala*	P + M	[18]	
	Chile	*D. uniflora*	M	[17]	
*Drosera* sect. *Drosera*	BR	*D. amazonica*	M	[22]	
	Colombia	*D. amazonica × biflora*	P, (M-tr)	[22]	
	Germany	*D. anglica*	M	[27,29,33]	(syn. *D. longifolia*)
	Germany	*D. anglica* × *rotundifolia*	M	[27]	(syn. *D. ×obovata*)
	Venezuela	*D. arenicola*	M	[18]	
	Venezuela	*D. biflora*	P	[22]	
	Venezuela	*D. biflora* × *esmeraldae*	P + M	[22]	
	BR	*D. brevifolia*	P, (M-tr)	[22]	
	USA	*D. brevifolia*	P?	[38]	
	?	*D. capillaris*	P	[30,31]	
	BR, USA	*D. capillaris*	M	[17]	
	?	“*D. communis*”	M	[32,33]	probably in err. (misidentified *D. spatulata* from cult.?)
	BR	*D. communis*	P	[18,38,39]	
	Venezuela	*D. esmeraldae*	M	[22]	
	Venezuela	*D. felix*	M	[17]	
	USA	*D. filiformis*	M	[14,27]	
	USA	*D. filiformis* subsp. *tracyi*	M	[26,30,33]	
	USA	*D. filiformis* var. *floridana*	M	[17]	
	Venezuela	*D. intermedia*	P	[22]	
	Germany	*D. intermedia*	P, (M-tr)	[14,26,27,30,31,33,40]	(*D. longifolia* auct. in err.)
	USA	*D. intermedia* × *filiformis*	P + M	[14]	(syn. *D. ×hybrida*)
	USA (introd. in Europe)	*D. intermedia* × *rotundifolia*	P + M	[14]	(syn. *D. ×eloisiana*, *D. ×belezeana* auct. in err.)
	Venezuela	*D. kaieteurensis*	0	[18]	
	USA	*D. linearis*	M	[17]	
	NC	*D. neocaledonica*	M	[17]	
	China	*D. oblanceolata*	M	[18]	
	Germany	*D. rotundifolia*	(P-tr), M	[14,27,29,30,31,33,40]	
	E Asia	*D. spatulata*	M	[18,26,27,30,31,32,33]	
	Palawan	*D. ultramafica*	P + M	[17]	
*Drosera* sect. *Ptycnostigma*	ZA	*D. admirabilis*	M	[18,32]	
	NE Namibia	*D. affinis*	P	[17]	
	ZA	*D. aliciae*	(P-tr), M	[26,27,30,31,33]	
	cult. (ZA)	*D. aliciae* × *capensis*	M	[27]	
	ZA	*D. burkeana*	M	[18,26,27,30,31,32,33]	
	ZA	*D. capensis*	(P-tr), M	[26,27,30,31,32,33,41]	
	ZA	*D. cistiflora*	(P-tr), M	[17,26,27,33]	
	ZA	*D. collinsiae*	M	[18,32,33]	0 in [27]
	ZA	*D. cuneifolia*	(P-tr), M	[18,26,27,33]	
	ZA	*D. dielsiana*	M	[29]	
	Zambia	*D. flexicaulis*	P	[22]	
	ZA	*D. glabripes*	(P-tr), (M-tr)	[42]	possibly in err.
	ZA	*D. hilaris*	M	[27]	
	Madagascar	*D. madagascariensis*	M	[22]	
	Madagascar	*D. madagascariensis*	(P-tr), M	[18,26,27,33,36]	(*D. ramentacea* auct. in err.)
	Zambia	*D. madagascariensis*	M	[22]	
	ZA	*D. natalensis*	(P-tr), M?	[41]	
	ZA	*D. nidiformis*	M	[18]	
	Zambia	*D. pilosa*	M	[22]	
	ZA	*D. ramentacea*	M	[18]	
	ZA	*D. rubrifolia*	0	[18]	
	ZA	*D. slackii*	P	[18,27]	
	ZA	*D. trinervia*	M	[26,27,33]	
	ZA	*D. venusta*	M	[32,33]	P in [27], probably in err.
*Drosera* sect. *Brasilianae*	BR	*D. camporupestris*	0	[18]	
	BR	*D. chrysolepis*	0	[18]	
	BR	*D. grantsaui* × *tomentosa*	M	[18]	(syn. *D. ×fontinalis*)
	BR	*D. graminifolia*	M	[22]	
	BR	*D. grantsaui*	M	[18]	
	BR	*D. graomogolensis*	0	[18]	
	BR	*D. latifolia*	P + M	[18]	
	BR	*D. magnifica*	M	[22]	
	BR	*D. montana*	P?	[38]	possibly in err.
	BR	*D. spiralis*	M	[18]	
	BR	*D. tentaculata*	M	[22]	
	BR	*D. villosa*	P + M	[18,27]	

^1^ Classification according to [43]; ^2^ AU = Australia; NT = Northern Territory (Australia); Qld. = Queensland (Australia); N.Guin. = New Guinea; NC = New Caledonia; ZA = South Africa; BR = Brazil; ^3^ P = Plumbagin (2-methyljuglone); M = Ramentaceone (7-methyljuglone); tr = trace; 0 = No naphthoquinone detected.

## Data Availability

All Figures are copyrighted by Jan Schlauer (Figure 1 and Figure 3) and Andreas Fleischmann (Figure 2 and Figure 4).

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
