# Peer review of "Distribution of Acetogenic Naphthoquinones in Droseraceae and Their Chemotaxonomic Utility"

_biology, 2024, doi:10.3390/biology13020097_

Round 1
Reviewer 1 Report
Comments and Suggestions for Authors
Dear Authors,
I hope this letter finds you well. I have carefully reviewed the manuscript titled “Distribution of Acetogenic Naphthoquinones in Droseraceae and their Chemotaxonomic Utility”. I appreciate the opportunity to review your work and provide feedback for its improvement. I believe that few corrections and suggestions for improvement could enhance the clarity and impact of your findings. Please consider the following points:
• Can you elaborate on the various biosynthetic routes mentioned for naphthoquinones, and how are they related to different classes of natural products?
• Add text too in “2. Review of Naphthoquinone Data from Droseraceae”
• Table 1 should be self-explanatory. Make it clearer to understand. Simplify comments in the comment section of table 1.
• What are the potential roles of naphthoquinones in carnivorous plants, and how does their presence or absence correlate with the loss or retention of carnivory in certain lineages?
• How stable is the presence/absence of acetogenic naphthoquinones (plumbagin and ramentaceone) within and between populations, and how does this stability contribute to their value as phytochemical markers?
• What are the main challenges or bottlenecks in further investigating underexplored species groups in Droseraceae for chemotaxonomic progress?
Comments on the Quality of English Language
The paper could benefit from addressing minor grammatical and punctuation errors, as well as simplifying complex sentences to enhance overall clarity and readability. I suggest a thorough review of the manuscript to ensure accurate grammar and punctuation. Additionally, consider breaking down longer sentences, particularly those spanning 5-6 lines, to improve reader understanding.
Reviewer 2 Report
Comments and Suggestions for Authors
While this paper demonstrates strong potential, there may be room for enhancement in certain areas to align it more closely with the publication criteria of this journal. Otherwise, I have some minor recommendations to increase the quality of your manuscript. Be careful with the writing and mistakes.
You must write the Objectives of this paper and Material and Methods.
There are two keywords repeated in the article title. The keywords are “Acetogenic Naphthoquinones” and the family “Droseraceae”. In order to increase the visibility of your paper I recommend changing these keywords. If you change them by other keywords, you will increase the probability that your paper could be found by future readers when they look for your paper in some databases like Scopus for example. If you repeat the same words in the article title and in keywords, less people could find your work. So, you must think about the visibility of your research.
Line 32. Just because the abstract is in the databases of the worldwide papers the acronym “TLC” must be explained into brackets. The abstract is the first thing that a reader finds in a database, so, you must explain as better as possible.
Line 42. Just follow the rules of this journal in order to publish this article. You must write “Figure 1” in capitals instead of “figure 1”. This is a very common tiny mistake in your whole manuscript, please, look for this mistake and fix it.
Line 60. Just follow the rules of this journal in order to publish this article. You must write “Figure 3” in capitals instead of “figure 3”. This is a very common tiny mistake in your whole manuscript, please, look for this mistake and fix it.
Line 82. Just follow the rules of this journal in order to publish this article. You must write “Figure 2” in capitals instead of “figure 2”. This is a very common tiny mistake in your whole manuscript, please, look for this mistake and fix it.
Line 82. The sentence is a little bit confusing, please, rephrase. I would write as follows: “Because: 1. Their structural affinities to the respective naphthoquinones…”. Or even you can write in different lines naming the ideas or points.
Because this is a scientific journal you must write the authors of the species.
Line 111. You must write in capital those letters which are used to build an acronym, so you must write “Thin Layer Chromatography (TLC)” instead of “thin layer chromatography (TLC)”. This tiny correction is for future readers. This is a very common little mistake that is very common in your whole paper.
Lines 119-120. You must write in capital those letters which are used to build an acronym, so you must write “High-Performance Liquid Chromatography (HPLC)” instead of “high-performance liquid chromatography (HPLC)”. This tiny correction is for future readers. This is a very common little mistake that is very common in your whole paper.
Line 120. You must write in capital those letters which are used to build an acronym, so you must write “Ultra-high-Performance Liquid Chromatography (UPLC)” instead of “ultra-high-performance liquid chromatography (UPLC)”. This tiny correction is for future readers. This is a very common little mistake that is very common in your whole paper.
Line 121. You must write in capital those letters which are used to build an acronym, so you must write “Nuclear Magnetic Resonance spectroscopy (NMR)” instead of “nuclear magnetic resonance spectroscopy (NMR)”. This tiny correction is for future readers. This is a very common little mistake that is very common in your whole paper.
Line 145. Just follow the rules of this journal in order to publish this article. You must write “Figure 2” in capitals instead of “figure 2”. This is a very common tiny mistake in your whole manuscript, please, look for this mistake and fix it.
If you write the locations of the species or a map your paper will increase its high quality.
Figure 4 is not in the main text. Please, you must follow the rules of the journal. Every Figure and Table must be in the text as well. Fix it.
Otherwise, you must write much more references in the text to the Figures and Tables, there is no problem to be redundant, but sometimes is very difficult to find this in the main text. Please, fix this tiny mistake.
Please, remember to look for all the acronyms in this manuscript and write its meaning in brackets. This will make the reading easier for future readers.
Otherwise, the authors adequately developed the Introduction, presenting the problems but you must write explicitly the objectives of this paper.
The are no methods in this manuscript.
The Discussion is well developed, and the data presented are correctly compared with other papers.
The authors are to be congratulated for the results obtained in this article.
Comments on the Quality of English LanguageThe English is good.
Reviewer 3 Report
Comments and Suggestions for Authors
The paper presents a review of the specific compounds present in the platns classified within Droseraceae family, wtith an amphasis on naphthoquinones. The valuable contribution is a comparative analyses of phylogenetic pattern and description of the presence of different compounds in similar taxa at family level. Based on detailed analyses of records, the author confirmed the that acetogenic naphthoquinones plumbagin and ramentaceone are valuable phytochemical markers, that helps i identification and classification of taxa within Droseraceae family.
Some of the specific comments and recommendations are following:
row 30 "since the dawn of mankind" - please change this text, the dawn of mankind is very extensive and long period on time. 20,000 30,000 years in the past.
Introduction: you refer to description of naphthoqinones, and mention the presence of these and similiar or other compounds in plants. Suggestion is to include one paragraph with short general description of Drocearaceae family (you mentioned in row 20 that it contain 260 recognized species to data), some botanical, taxonomical and ecological information are required.
Chater 2: Review of Napthhoquinone Data from Droseraceae
You inserted Table 1., without previous sentence that is link to data showed in Table 1. It is important to write some text above the table.
row 197: Figure 3, please refer to Figure 3 in text
row 256: Figure 4, please refer to Figure 4 in text
chapter References: please check that You use the journal abbreviation in the whole list of references, because it is requested by the Journal Instructions to Authors. In some references, the journal names are not abbreviated (references numbers: 2, 5, 6, 7, 10, 24, 44...)
row 438, reference 54 needs rearrangement to "Phytochemistry 1986, 25, 1377-1383
The English Language is of good quality.
